# Unraveling the Nexus: Exploring the Relationship between Exercise Habits and Sleep Quality in Older Adults

**DOI:** 10.3390/healthcare11202759

**Published:** 2023-10-18

**Authors:** Wenhu Xu, Jianze Fang, Long Chen, Dongmin Wang, Chengye Huang, Tiange Huang, Chao Guo

**Affiliations:** 1Institute of Population Research, Peking University, Beijing 100091, China; 2101213176@stu.pku.edu.cn (W.X.); chenlong@stu.pku.edu.cn (L.C.); chengye.huang@stu.pku.edu.cn (C.H.); 2101213178@stu.pku.edu.cn (T.H.); chaoguo@pku.edu.cn (C.G.); 2School of Business, Macau University of Science and Technology, Macau 999078, China; 2109853gbo20007@student.must.edu.mo; 3Department of Physical Education, Peking University, Beijing 100091, China

**Keywords:** exercise habits, the quality of sleep, older adult

## Abstract

The objective of this study is to investigate the impact of exercise habits on enhancing the sleep quality of older adults. The Pittsburgh Sleep Quality Scale and other questionnaires were utilized to assess the sleep quality of older adults aged 60 years and above in the H district. The sampling method employed was stratified random sampling. To analyze the influencing factors of sleep quality, the average treatment effect was estimated, the robustness of the results was assessed and statistical methods such as Logit regression and propensity score matching were employed. The study revealed that exercise habits strongly correlated with improved sleep quality in the older adult (*p* < 0.05), with the average total sleep quality score being 6.22 (±3.53). It was observed that older adults who engaged in exercise habits experienced a significant 12.66% increase in the likelihood of achieving good sleep. This investigation highlights the positive association between exercise habits and enhanced sleep quality among older adults. Additionally, age, physical pain and self-rated health statuses were identified as significant factors influencing sleep quality in this population. To enhance the sleep quality of older adults, this article recommends promoting relevant exercise habits, thus contributing to their overall well-being and quality of life.

## 1. Introduction

Health is an inevitable requirement to promote the comprehensive development of exhibitions, and physical exercise is an activity consciously carried out by people to affect their health. However, with the development of the economy and society, the aging of the population and the change in living behavior have brought challenges to the maintenance and promotion of people’s health [1]. In the compendium for the construction of physical powerhouses issued by the State Council in 2019, the strategic task of “implementing the national strategy for universal fitness and powering up health China Construction” is proposed, which states that physical activity should be developed for key populations such as the older adult, which make them enjoy the implementation strategy for corresponding physical health intervention programs [2]. Sleep disorder is a common phenomenon in the life of the older adult. Due to the increasing degree of aging in China, sleep disorders have become an important factor threatening the physical and mental health of the older adult. More and more older adults will face sleep disorders, which will affect the treatment and rehabilitation of their primary illnesses and increase the likelihood of physical illnesses [3]. Previous studies have focused on different health conditions [4,5] or explored the relationship between different intervention methods [6,7] and sleep quality in the older adult. Based on the existing body of literature, the predominant focus among scholars has been on employing specific intervention methods to impact an individual’s sleep quality. However, there is a noticeable dearth of studies in fields that have undertaken correlation analyses involving enduring exercise routines as a pivotal variable influencing one’s overall sleep health status.

Most studies related to the effects of exercise habits on sleep quality at home and abroad have focused on the child and adolescent domains [8,9,10,11,12], with a limited number of research studies addressing the older adult domain [13,14,15]. Most of the studies believe that exercise habits positively affect sleep quality. Hence, when considering age stratification, it is evident that there is a paucity of comprehensive investigations delving into the intricate nexus between exercise routines and sleep quality within the older adult.

In light of pertinent policy directives issued by the Chinese government, which underscore the importance of harmonizing physical well-being and overall health [16], this research endeavor seeks to align with the guidelines set forth in the Healthy China Action (2019–2030). The primary objective of this study is to explore the intricate relationship between exercise patterns and sleep quality among the elderly population in China. By autonomously collecting data in accordance with the stipulated standards, this study aims to furnish a theoretical foundation for the development of exercise-based interventions for older adults grappling with sleep quality disorders.

## 2. Methods

Existing studies have shown that when conducting research on sleep quality in the older adult, most observations are made from three dimensions. First, most studies confirm the influence of basic demographic variables such as gender, age, education level, and marital status on the sleep status of the older adult [17,18,19]. Most scholars have included factors such as family support or social support in their studies when analyzing the influencing factors of sleep quality [16,20]. In addition, different types of chronic diseases, skin itching, and the self-assessment of health statuses have also been mentioned by scholars many times in their studies [21,22,23,24,25,26]. As with other studies, drinking can be presented as categorical variables in the study of the health status of older adults [27].

### 2.1. Data Sources

The source of data for this study is the cross-sectional data collected autonomously by the project team in the project carried out in district H. The study collected microdata, including information on exercise habits, demographic variables, family exercise support, and physical health status of the participants during the period from October 2020 to November 2020. On the one hand, District H, as the largest economy under the jurisdiction of Beijing, holds significant economic and social importance. On the other hand, according to data, the resident population of district H is 3,133,469, of which 18.46% are aged 60 and above, surpassing the national average and approaching the overall aging rate of Beijing. This demographic composition makes District H highly representative in terms of population structure. Hence, during the period from October 2020 to January 2021, this study established the sampling proportion by calculating the percentage of the six communities in Y district relative to the total population of the selected communities. Taking note of the minimum representative numbers, approximately 4% of the target population was randomly selected as a sample group. Subsequently, the researchers conducted stratified random sampling with the older adult population from communities under Y street, namely CC, WX, ZG, YB, CZ, and YD, as the research subjects. The inclusion criteria required participants to be aged 60 or above and be local permanent residents. After providing preliminary research training to the interviewers, face-to-face computer-assisted personal interviews (CAPI) were carried out with the participants. Ultimately, the study included 370 subjects in its analysis.

### 2.2. Instruments and Variables

The Pittsburgh Sleep Quality Index (PSQI) scale was primarily designed to assess the sleep quality of the study subjects over the past month. It comprises seven components: subjective sleep quality, sleep latency, sleep duration, habitual sleep efficiency, sleep disturbance problems, sleep medication use, and daytime dysfunction. Each component is scored on a scale of 0 to 3. The total PSQI score is obtained by summing the scores of each component, resulting in a total score ranging from 0 to 21. Higher scores indicated worse nocturnal sleep quality.

After multiple rounds of verification and discussion, the question “whether or not you exercise for 30 min or more three times a week” was refined and changed to “whether or not you have an exercise habit.” In other words, based on the “yes” or “no” answers, the older adults were categorized as either “having an exercise habit” or “not having an exercise habit” if they consistently engaged in exercise three times a week or more, with each session lasting 30 min or more.

Dependent variables: The reliability and validity of the Pittsburgh Sleep Quality Index in China have been established through previous literature reviews and reference to appropriate testing [28,29]. This study draws upon research findings from Chinese scholars and employs a score of 7 as the threshold for distinguishing individuals with sleep disorders from those without sleep disorders [30,31]. Dichotomous variables were then set to determine whether the final PSQI index score exceeded 7. A score of 1 indicated a sleep quality score of less than or equal to 7 points, signifying good sleep quality. Conversely, a score of 0 denoted a sleep quality score greater than 7 points, indicating poor sleep quality.

Independent variables: This study carefully defined the concept of exercise habits by reviewing pertinent literature and adopting the fundamental definition of exercise habits. The relevant standards from the “Healthy China Action” (2019–2030), advocating for more than 30 min of moderate-intensity exercise at least three times a week, were also taken into consideration as a reference for defining exercise habits in the context of this research [32].

Control variables: This study conducted a thorough review of relevant literature both domestically and internationally to identify factors that may influence the sleep quality of older adults, as supported by existing research studies. Subsequently, one-way tests were performed for demographic variables, family support variables, health status variables, and sleep quality. Variables that demonstrated significance levels of 10% or less were included in the model as control variables. Based on the results presented in Table 1, the following types of variables were selected as control variables for determining whether the older adult had sleep disorders: (1) basic demographic variables, which included age indicators according to the one-way test criteria; (2) family support variables, including whether they lived alone, whether the number of living children was less than two, and whether the mother participated in sports regularly; and (3) health status-related variables, including whether they had pain, whether they had chronic diseases, whether they had pruritus, and self-rated health statuses.

### 2.3. Data Analysis

This study employed Stata 16.0 software for data analysis. Firstly, a descriptive analysis of variables was conducted to provide an overview of the data. Subsequently, a one-way analysis of variables was performed using the χ2 test to screen for relevant variables. Next, the study constructed three separate logit regression models, gradually incorporating selected variables for analysis. To estimate the relationship between exercise habits and sleep quality, Propensity Score Matching (PSM) was employed. Additionally, a robustness test was conducted to ensure the reliability of the findings. In the analysis, a significance level of *p* < 0.05 was considered statistically significant, indicating a notable difference between the variables. The use of Propensity Score Matching aimed to mitigate endogeneity problems arising from sample selection bias and individual heterogeneity. By addressing these biases, the study sought to achieve more accurate estimates of the factors influencing sleep quality in older adults. The mean treatment effect and the goodness of fit of the regression model were evaluated to further validate the findings.

## 3. Results

### 3.1. Basic Situation of Sleep Disorders in People with Different Characteristics

Among the 370 older adults included in this study, 59.19% were aged 60 to 69 years, 29.73% were aged 70 to 79 years, and 11.08% were aged 80 years and older, with an average age of 68.74 years. Moreover, the majority of older adults (89.46%) were not living alone. For further details on the other variables and the prevalence of sleep disorders among individuals with different characteristics, please refer to Table 1, which provides a comprehensive overview of the relevant data in this study.

In the surveyed population, it was observed that 75.75% of respondents who maintained regular exercise routines fell within the category characterized by the absence of sleep disorders, while the remaining 24.25% belonged to the cohort exhibiting symptoms of sleep disorders. Specifically, this implies that 14.29% of individuals within the non-sleep disorder group did not engage in regular exercise, whereas 29.81% of those within the sleep disorder group were devoid of exercise habits.

### 3.2. Basic Information about the Quality of Sleep

This study presents a descriptive analysis of the Pittsburgh Sleep Quality Index (PSQI) status among 370 older adults. Based on the existing literature review, it is common in domestic research to use a PSQI score greater than 7 as the cut-off to assess the sleep quality status of respondents.

The PSQI scores for each factor were as follows: sleep quality scored 0.97 ± 0.75, time to fall asleep scored 1.36 ± 0.56, sleep duration scored 1.07 ± 0.99, sleep efficiency scored 0.60 ± 0.99, sleep disturbance scored 1.12 ± 0.53, use of sleeping pills scored 0.32 ± 0.81, and daytime functioning scored 0.78 ± 0.88.

A breakdown of the results indicated that 70 people (18.92%) had a sleep quality score greater than 1, 120 people (32.43%) scored above 1 for time to fall asleep, 114 people (30.81%) scored above 1 for sleep duration, 75 people (20.27%) scored above 1 for sleep efficiency, 74 people (20.00%) scored above 1 for sleep disturbance, 33 people (8.92%) scored above 1 for the use of sleeping medication, and 64 people (17.30%) scored above 1 for daytime functioning.

The mean overall sleep quality score was 6.22 (±3.53), with 104 participants scoring above 7. Thus, the proportion of older adults in the region with poor sleep quality was 28.11%.

### 3.3. Regression Analysis of the Sleep Quality Profile of Older Adults

This study employed a stepwise approach to control for confounding variables, including demographic variables, family exercise support variables, and own health status variables. The first model included only basic demographic variables, the second model incorporated family exercise support variables, and the third model further added health-status-related variables. The specific representation of the models is as follows:*Logit* (Y*_i_*) = *β*_0_ + *β_i_ Dem_i_* + *e*_1_
(1)

*Logit* (Y*_i_*) = *β*_0_ + *β_i_ Dem_i_* + *β_j_ Fam_j_* + *e*_2_
(2)

*Logit* (Y*_i_*) = *β*_0_ + *β_i_ Dem_i_* + *β_j_ Fam_j_* + *β_k_ Hea_k_* + *e*_3_
(3)


As illustrated in Equations (1)–(3), Y*_i_* represents the probability that the *i*th older adult has a sleep disorder. *Dem_i_* corresponds to the underlying demographic variable, *Fam_j_* denotes the family exercise support variable, and *Hea_k_* represents the health status variable. The regression coefficients for the independent variables, *β_i_*, *β_j_*, and *β_k_*, reflect the extent of influence each variable has on the dependent variable while considering the influence of other variables. Additionally, the error term denoted as “*e*”, accounts for unexplained variations in the model.

The final pseudo-R2 value for Model 3 is 12.20%, indicating that the model explains 12.20% of the variance in sleep quality among older adults.

Based on the results presented in Table 2, after controlling for different types of variables multiple times, the logit regression model tests revealed that the odds ratio (OR) of sleep quality remained consistent at 2.09 (with a 95% confidence interval of 1.12 to 3.88) for older adults with exercise habits compared to those without exercise habits. This result was statistically significant (*p* < 0.05). In practical terms, it means that older adults with exercise habits were 2.09 times more likely to have better sleep quality compared to those without exercise habits. Furthermore, for each unit increase in the value of the exercise habit variable in older adults, the odds of having better sleep quality would increase by a factor of 2.09.

By calculating the average marginal effect, it was determined that every time an older person adopts an exercise habit, the probability of having better sleep health increases by 12.66%.

In Model 3, three variables demonstrated a statistically significant relationship with the dependent variable. The odds ratio for sleep quality was 0.60 times higher for older adults in the higher age group compared to those in the lower age group, indicating that older adults in the higher age group were less likely to have better sleep quality than those in the lower age group. The odds ratio for sleep quality was 0.41 times higher for older adults with physical pain conditions compared to those without physical pain conditions, suggesting that older adults with physical pain were less likely to have good sleep quality. Conversely, compared with those with poor self-rated health, the odds ratio for sleep quality was 3.77 times higher in older adults with better self-rated health, indicating that older adults with better self-rated health were more likely to have better sleep quality than those with poor self-rated health.

### 3.4. Robustness Tests

In this study, the propensity score matching method was employed to address potential endogeneity problems arising from sample selectivity bias. This approach aimed to investigate the relationship between exercise habits and sleep quality in older adults more rigorously. By estimating the average treatment effect of having an exercise habit on sleep quality, the study sought to reduce biases that might exist due to pre-sampling, sample selectivity bias, and the influence of confounding variables. This allowed for a more accurate estimation of the factors influencing sleep quality in older adults and the overall intervention outcome of the regression model.

To ensure the robustness of the average treatment effect test, Logit regression was conducted following various matching methods, such as one-to-one nearest neighbor matching, one-to-four nearest neighbor matching, caliper matching and kernel matching. Table 3 presents a graphical representation of the sleep quality status of the older adult after matching the propensity scores for the presence or absence of exercise habits. This analysis was performed to enhance the validity and reliability of the study’s findings.

This paper employs the propensity score matching method to address the endogeneity problem and estimate the mean treatment effect of a regression model with exercise habits as the core independent variable. Robustness tests are also conducted to validate the findings.

The propensity score matching method proves effective in alleviating the endogeneity problem commonly associated with ordinary logit regression models based on cross-sectional data. After multiple matching, the Average Treatment Effect on the Treated (ATT) scores indicate that the difference between the observed results of individual respondents in the intervention group and their counterfactuals falls between −0.16 and −0.13. This suggests that individuals who developed exercise habits demonstrated improved sleep quality compared to the group that did not develop exercise habits.

Furthermore, the improvement in ATT scores after multiple matching indicates that the average treatment effect is enhanced by nearest neighbor matching, caliper matching, and kernel matching. Importantly, the model’s pseudo-R2 remains relatively unchanged after multiple matching.

Logit regression models also demonstrate a significant relationship between exercise habits and sleep quality. Consequently, this study concludes that the presence or absence of exercise habits among older adult in the communities under Y street is significantly associated with their sleep quality.

## 4. Discussion

This study scrutinized the prevailing sleep quality among older adults and systematically investigated the correlation between their exercise habits and sleep quality. It was observed that older adults who engaged in exercise habits experienced a significant increase in the likelihood of achieving good sleep. The primary data collected during the preliminary survey were carefully analyzed by the project team, and launched the following discussion on the results:

First of all, the sleep disorder rate of the older adult collected in this survey is between the survey data of Liu Lianqi et al. (11.1%) based on Shandong [25] and the survey data of Liu Haijuan based on 22 provinces of China (49.9%) [28], accounting for about 28.11%. The local older adult with poor sleep quality maintains the normal range which accords with the actual situation in China [33]. It is essential to emphasize that the data collected in this study were mainly collected during the COVID-19 period. In comparison to studies on sleep quality among older adults in countries such as the United States and Italy, where the novel coronavirus pandemic has been prevalent [34,35], the findings of this study reveal that the surveyed older adults exhibit a relatively low PSQI score, indicating a relatively high level of sleep quality. However, it is imperative to acknowledge that this research cannot conclusively exclude the potential influence of the public health emergency caused by COVID-19 on the sleep quality of the older adult. This phenomenon may be attributed to the fact that, much like the mechanism observed in hospitalization cases [36], acute sleep deprivation resulting from COVID-19 prevention and control measures can also alter the living environment, potentially resulting in diminished sleep quality.

Secondly, after controlling for confounding variables, this study believes that the exercise habits of the older adult can effectively affect their sleep quality. This means that older adults with exercise habits are less likely to have sleep disorders than those without exercise habits. It can be observed that exercise habits are protective factors for the sleep quality of the older adult, which is consistent with the positive impact of exercise habits described in several foreign studies on adolescents [12,14]. This shows that exercise habits have a certain effect on the care of different age groups. Drawing from the initial interview findings, this study discerned that the primary physical activities engaged in by the participants within the community encompassed practices such as the Eight Section Brocade, Five Animals of Qigong, Tai Chi Chuan, and walking, among others. Nevertheless, owing to the absence of specific parameters delineating the various forms of exercise within the questionnaire, the study encountered challenges in adequately capturing and categorizing the exercise modalities undertaken by the respondents, thereby impeding the prospects for more in-depth research in this regard. A study conducted in South Korea demonstrated a substantial influence of varying sleep patterns on the regular exercise habits of the elderly, suggesting a potential causal connection between exercise routines and sleep quality [37]. However, it is important to note that this particular study was unable to establish a clear mechanism explaining how exercise habits directly affect the sleep quality of older individuals.

In addition, some demographic variables, family exercise support variables and health status were significantly associated with sleep quality in the older adult. People aged 80 and over are more likely to have poor sleep quality than those aged 60 to 69, which is consistent with previous studies showing that sleep quality deteriorates with age [23,24,38,39]. Compared with the older adult without physical pain, the older adult’s physical pain significantly affects their sleep, which can also respond to previous research results [40]. Compared with those with poor self-rated health status, the older adult with better self-rated health statuses have better sleep quality, indicating that when older adults have better self-rated health status, the probability of their good sleep status is correspondingly greater, which again proves that the older adult’s good self-rated health status can affect their sleep status [41]. However, it is worth mentioning that while the overall sleep quality score serves as an important indicator for assessing the sleep quality of individuals across various age groups, it is pertinent to note that, in the context of other research avenues focusing on the sleep health of the older adult, sleep duration represents a variable of considerable breadth [42,43]. Despite often being regarded as the primary influencing factor, it should not preclude its consideration for comparative analysis in studies examining sleep outcomes. While this emphasis aligns with the prominence of the CHARLS database in Chinese academic circles, the association between exercise habits and sleep duration warrants further in-depth exploration in future research endeavors.

## 5. Conclusions

In conclusion, this study initially adopts the pertinent criteria outlined in the Healthy China Initiative (2019–2030) to define exercise habits. The ultimate findings indicate that the sleep quality of the elderly in this region falls within the spectrum observed in Chinese adults. Concurrently, the study reveals a robust correlation between exercise habits and the sleep quality of elderly individuals. However, it is essential to delve deeper into the potential repercussions of the outbreak for a more comprehensive understanding. 

Secondly, the age of the older adult, the presence of physical pain and their self-rated health status emerge as pivotal factors influencing their sleep quality. Consequently, community health management should place a strong emphasis on understanding individuals’ health perspectives and fostering tailored exercise habits. In contrast to traditional exercise interventions, which are rooted in conventional research paradigms, this paper advocates for the comprehensive development of enduring exercise routines among the elderly population.

Lastly, while this study did not delve into a detailed mechanism analysis, a review of other literature sources [44,45,46,47] reveals that exercise habits can exert a positive influence on various physical conditions among the elderly by enhancing physical function. Consequently, this study recommends that policymakers define the essence of exercise habits and underscore the significance of exercise duration, drawing upon the insights gleaned from the literature comparisons. This approach can serve as a valuable foundation for establishing effective criteria to promote physical well-being in the elderly and formulate exercise prescriptions related to sleep quality in future endeavors.

## Figures and Tables

**Table 1 healthcare-11-02759-t001:** Basic demographic variables and basic information about sleep disorders in people with different characteristics (*n* = 370).

Features	Sample Size	Sleep Non-Disordered Group	Sleep Disorder Group	χ^2^	*p*-Value
Age				14.050 ***	0.001
60–69 years old	219 (59.19%)	169 (77.17%)	50 (22.83%)		
70–79 years old	110 (29.73%)	77 (70.00%)	33 (30.00%)		
80 years old and above	41 (11.08%)	20 (48.78%)	21 (51.22%)		
Living alone or not				3.600 *	0.058
No	331 (89.46%)	243 (73.41%)	88 (26.59%)		
Yes	39 (10.54%)	23 (58.97%)	16 (41.03%)		
The mother participated in sports regularly in the early years				3.317 *	0.069
No	294 (79.46%)	205 (69.73%)	89 (30.27%)		
Yes	76 (20.54%)	61 (80.26%)	15 (19.74%)		
Number of children within 2				3.065 *	0.080
No	246 (66.49%)	184 (74.80%)	62 (25.20%)		
Yes	124 (33.51%)	82 (66.13%)	42 (33.87%)		
Pain in body				15.500 ***	0.000
No	167 (45.14%)	137 (82.04%)	30 (17.96%)		
Yes	203 (54.86%)	129 (63.55%)	74 (36.45%)		
Body itching				5.194 **	0.023
No	276 (74.59%)	207 (75.00%)	69 (25.00%)		
Yes	94 (25.41%)	59 (62.77%)	35 (37.23%)		
Chronic disease				6.259 **	0.012
No	98 (26.49%)	80 (81.63%)	18 (18.37%)		
Yes	272 (73.51%)	186 (68.38%)	86 (31.62%)		
Good health status				19.230 ***	0.000
No	26 (7.03%)	9 (34.62%)	17 (65.38%)		
Yes	344 (92.97%)	257 (74.71%)	87 (25.29%)		
Drinking or not				11.874 ***	0.001
No	69 (18.65%)	38 (55.07%)	31 (44.93%)		
Yes	301(81.35%)	228 (75.75%)	73 (24.25%)		
Have exercise habits				11.874 ***	0.001
No	69 (18.65%)	38 (55.07%)	31 (44.93%)		
Yes	301 (81.35%)	228 (75.75%)	73 (24.25%)		
Total	370 (100%)	266 (71.89%)	104 (28.11%)		

Note: (1) ***, **, and * indicate significant at the 1%, 5%, and 10% statistical levels, respectively; (2) Values in parentheses indicate frequencies; (3) Chinese scholars tested the reliability and validity of this scale, in which Cronbach’α coefficient was 0.842 and retest reliability was 0.809 which could be used to evaluate the sleep quality of the Chinese population, and 7 was divided into PSQI critical value.

**Table 2 healthcare-11-02759-t002:** Logit regression analysis model of exercise habits and sleep status of the older adult.

Variable	(1) OR (95% CI)	(2) OR (95% CI)	(3) OR (95% CI)
Age	0.59 *** (0.43–0.83)	0.55 *** (0.35–0.84)	0.59 ** (0.36–0.95)
Living alone or not		0.60 (0.29–1.24)	0.49 ** (0.23–1.06)
Does mother regularly engage in sports		1.78 * (0.91–3.48)	1.88 (0.91–3.89)
Alive children 2 or less		1.39 (0.72–2.68)	1.14 (0.56–2.35)
Pain in body			0.41 *** (0.24–0.71)
Chronic disease			0.82 (0.44–1.56)
Body itching			0.68 (0.40–1.18)
Physical health self-assessment			3.90 *** (1.70–8.92)
Drinking or not			1.11 (0.83–1.48)
Have exercise habits	2.34 *** (1.33–4.15)	2.38 *** (1.34–4.22)	2.09 ** (1.12–3.88)

Note: Heteroskedasticity robust standard error intervals are in brackets, ***, ** and * indicate significance at 1%, 5% and 10% statistical levels, respectively.

**Table 3 healthcare-11-02759-t003:** PSM-Logit robustness test for the presence or absence of exercise habits and sleep quality status of older adults.

Variable	(5)	(6)	(7)	(8)
Age	0.60 (0.32–1.13)	0.53 ** (0.30–0.95)	0.60 ** (0.26–1.39)	0.58 ** (0.35–0.95)
Living alone or not	0.23 ** (0.53–1.00)	0.30 ** (0.10–0.92)	0.60 (0.26–1.39)	0.50 ** (0.23–1.10)
Does mother regularly engage in sports	2.12 (0.49–9.21)	3.14 ** (1.01–9.79)	1.80 (0.81–3.97)	1.92 (0.85–4.33)
Alive children 2 or less	1.45 (0.52–4.01)	1.65 (0.69–3.93)	1.20 (0.56–2.55)	1.15 (0.55–2.42)
Pain in body	0.38 *** (0.19–0.78)	0.35 *** (0.18–0.68)	0.89 *** (0.46–1.72)	0.42 *** (0.24–0.73)
Chronic disease	0.77 (0.28–2.14)	0.77 (0.30–1.96)	0.89 (0.46–1.72)	0.88 (0.45–1.70)
Body itching	0.79 (0.37–1.69)	0.76 (0.38–1.53)	0.67 (0.38–1.19)	0.65 (0.37–1.14)
Physical health self-assessment	3.79 ** (1.28–11.25)	4.95 *** (3.01–12.17)	3.87 *** (1.58–9.47)	3.94 *** (1.72–9.01)
Drinking or not	0.83 (0.48–1.43)	0.90 (0.36–16.07)	0.98 (0.70–1.38)	0.98 (0.70–1.38)
Have exercise habits	2.61 *** (1.26–5.41)	2.70 *** (1.38–5.29)	2.15 ** (1.15–4.04)	2.12 ** (1.14–3.97)

Note: Heteroskedasticity robust standard errors are in brackets, ***, ** indicate significance at the 1%, and 5% statistical levels, respectively.

## Data Availability

The data provided in this research can be obtained by reaching out to the corresponding author upon request. The data cannot be accessed publicly, as it is restricted due to considerations related to privacy and ethical concerns.

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
