# Peer review of "Unraveling the Nexus: Exploring the Relationship between Exercise Habits and Sleep Quality in Older Adults"

_healthcare, 2023, doi:10.3390/healthcare11202759_

Round 1
Reviewer 1 Report
I really appreciate the topic taken up by the authors.
The introduction, the methodological part and the description of the research results have been prepared in a sufficient way.
The discussion needs elaboration. It would also be worth referring to studies outside of China to determine whether there are significant differences between different regions of the world.
In conclusions, please refer directly to the research problem of this article. It would be worth expanding them further, as well as expanding them with the part concerning the limitations of the research carried out.
Author Response
Thank you for bringing this to our attention. We concur with all of your comments. In response, we conducted a cross-regional analysis of the sleep quality among the elderly participants surveyed by our research institute. In lines 296 to 297, we compared the variations in sleep quality among the elderly during the COVID-19 pandemic and emphasized the significance of acknowledging the COVID-19 pandemic as a global public health emergency. Consequently, it is important to acknowledge that this study does not delve into the mechanisms through which the COVID-19 pandemic affects the elderly population, as highlighted in lines 303 to 305.

Reviewer 2 Report
Τhe manuscript deals with the relationship between exercise habits and sleep quality in older adults. This is an interesting and significant study. However, I have some methodological questions and comments, which I would like to be addressed.
Minor comments:
Authors have used subjective methods of assessing exercise habits and sleep quality. A high percentage of the population studied said that they were drinking (what was the amount?). Were subjects in a mental stage capable of answering the questions reliably?
Methods
Add sex of the population studied and their socioeconomic /educational status.
Please provide more info about the PSQI scoring, as: a. higher scores indicated worse nocturnal sleep quality. b. the period studied (for example: over a one-month time interval?)
Please explain why you have chosen PSQI score greater than 7 as the cut-off to assess the sleep quality status of respondents. (Lines 160-161). The last 30 years all studies use the cutoff score of 5 for the PSQI global score introduced by Buysse and colleagues (Buysse et al., Citation1989) to discriminate between “poor” sleepers and “good” sleepers.
Why have you chosen to use a simple questionnaire for physical activity habits and not validated questionnaires on physical activity, as IPAQ, etc.
Results
Please refer only to the results in this section. The explanation and meaning/significance of the results should be removed to the discussion.
Discussion
Please compare the results of the present study with other previous studies worldwide. Authors have used only studies from China. Please, see what in other countries exist and also add international references.
Add the limitations of the study.
Author Response
Unraveling the Nexus: Exploring the Relationship Between Exercise Habits and Sleep Quality in Older Adults
|
Response to Reviewer 2 Comments
|
||
|
1. Summary |
|
|
|
Thank you very much for taking the time to review this manuscript. Please find the detailed responses below and the corresponding revisions in track changes in the re-submitted files. |
||
|
2. Questions for General Evaluation |
Reviewer’s Evaluation |
Response and Revisions |
|
Does the introduction provide sufficient background and include all relevant references? |
Yes |
|
|
Are all the cited references relevant to the research? |
Can be improved |
We have broadened the existing literature by engaging in cross-regional discussions centered on our findings. |
|
Is the research design appropriate? |
Can be improved |
We supplemented the research design section. |
|
Are the methods adequately described? |
Can be improved |
We explain the shortcomings in the method description. |
|
Are the results clearly presented? |
Can be improved |
We expanded our findings. |
|
Are the conclusions supported by the results? |
Yes |
|
|
3. Point-by-point response to Comments and Suggestions for Authors |
||
|
Comments 1: Τhe manuscript deals with the relationship between exercise habits and sleep quality in older adults. This is an interesting and significant study. However, I have some methodological questions and comments, which I would like to be addressed. Methods (1)Add sex of the population studied and their socioeconomic /educational status. Please provide more info about the PSQI scoring, as: a. higher scores indicated worse nocturnal sleep quality. b. the period studied (for example: over a one-month time interval?) (2)Please explain why you have chosen PSQI score greater than 7 as the cut-off to assess the sleep quality status of respondents. (Lines 160-161). The last 30 years all studies use the cutoff score of 5 for the PSQI global score introduced by Buysse and colleagues (Buysse et al., Citation1989) to discriminate between “poor” sleepers and “good” sleepers. (3) Why have you chosen to use a simple questionnaire for physical activity habits and not validated questionnaires on physical activity, as IPAQ, etc. (4) Results:Please refer only to the results in this section. The explanation and meaning/significance of the results should be removed to the discussion. (5) Discussion:Please compare the results of the present study with other previous studies worldwide. Authors have used only studies from China. Please, see what in other countries exist and also add international references. (6)Add the limitations of the study. |
||
|
Response 1: Thank you sincerely for your invaluable review and the meticulous suggestions you have provided. We deeply appreciate all of your recommendations. Below is our response: (1) During the correlation analysis, we observed that variables such as gender, socioeconomic status, and education level did not exhibit a significant relationship with the dependent variables. Consequently, they were not incorporated into the regression model. Additional information pertaining to the PSQI scores, including details about the study period and the PSQI scale, has been provided in lines 108-114, 91-92, and 114 for reference. (2) Following the introduction of the PSQI questionnaire in China, several scholars conducted reliability and validity tests to assess its applicability. Subsequently, in Chinese research, a score of 7 has been employed as a consistent threshold to categorize the sleep status of Chinese respondents. This standard continues to be utilized in recent Chinese studies, as elaborated in lines 123 to 125. (3) We sincerely appreciate your guidance. Regrettably, in our previous attempts, we used the Chinese version of the IPAQ long questionnaire to gather data from elderly participants. However, the extended time required for respondents to answer this section led the interviewers on our project team to express concerns regarding the reliability of the responses. After careful deliberation, we have made the decision not to incorporate the results from the IPAQ into our data processing for this study. (4) In the discussion section, we have engaged in a comprehensive analysis, incorporating feedback from other reviewers to provide a more expansive examination of the results. (5) Thank you for bringing this to our attention. In response, we conducted a cross-regional analysis of the sleep quality among the elderly participants surveyed by our research institute. In lines 296 to 297, we compared the variations in sleep quality among the elderly during the COVID-19 pandemic and emphasized the significance of acknowledging the COVID-19 pandemic as a global public health emergency. Consequently, it is important to acknowledge that this study does not delve into the mechanisms through which the COVID-19 pandemic affects the elderly population, as highlighted in lines 303 to 305. (6) We have incorporated discussions regarding the limitations into each respective section of the discussion, and we sincerely value your input. These discussions can be found in Lines 294-305, Lines 316-324, and Lines 337-339. |
||
|
Comments 2: Minor comments: (1) Authors have used subjective methods of assessing exercise habits and sleep quality. (2) A high percentage of the population studied said that they were drinking (what was the amount?). (3) Were subjects in a mental stage capable of answering the questions reliably?
|
||
|
Response 2: Thank you for the reviewer's valuable comments. Our responses to the minor comments are outlined below: (1) As previously mentioned, upon assessing the gathered data, it is imperative that we should discard the data obtained from the IPAQ questionnaire. We will proceed with the analysis of the exercise habits among the elderly using the following approach: Referencing the "Healthy China 2030" Outline published by the CPC Central Committee and the State Council, we will focus on the specific criterion of whether individuals engage in physical activity for a minimum of 30 minutes, at least three times a week. (2) The alcohol consumption measurement standard employed in this study was derived from the relevant measurement method found in the CHARLS database, which categorizes consumption by frequency. Given the infrequent nature of alcohol consumption reported in the study (e.g., once a week or less), we aggregated variables below 5% into a single category labeled as "yes" (indicating alcohol consumption). Subsequently, this allowed us to create a dichotomous variable to determine whether individuals were drinkers or non-drinkers. (3) Our interviewers underwent comprehensive professional training before conducting face-to-face computer-assisted personal interviews (CAPI) with the participants. The interviewers evaluated the interview process and the respondents' conditions. Subsequently, they recorded the responses in the data collection terminal, ensuring the overall data credibility. |
||
|
|
||
|
|
||
|
|
||
|
|
||
|
|
||

Reviewer 3 Report
First at all, Is a good job, I think is concise in the information, easy to read. I have three thoughts about your work. The first one is to have a better description of what has been found in relation to sleep habits in adults, since they only point out that there are few studies, but a better description would highlight their introduction. The second thing is abourt exercise habits. Same thing, I would like to know what tipe of exercise the older adults do. Is not explained in the data collection. It would be interesting. The last thing for discussion is to ask how to get more adults to exercise? It would be interesting know theirs opinions.
Author Response
Unraveling the Nexus: Exploring the Relationship Between Exercise Habits and Sleep Quality in Older Adults
|
Response to Reviewer 3 Comments
|
||
|
1. Summary |
|
|
|
Thank you very much for taking the time to review this manuscript. Please find the detailed responses below and the corresponding revisions in track changes in the re-submitted files. |
||
|
2. Questions for General Evaluation |
Reviewer’s Evaluation |
Response and Revisions |
|
Does the introduction provide sufficient background and include all relevant references? |
Can be improved |
We have added to the introduction. |
|
Are all the cited references relevant to the research? |
Yes |
|
|
Is the research design appropriate? |
Yes |
|
|
Are the methods adequately described? |
Yes |
|
|
Are the results clearly presented? |
Yes |
|
|
Are the conclusions supported by the results? |
Yes |
|
|
3. Point-by-point response to Comments and Suggestions for Authors |
||
|
Comments 1: First at all, Is a good job, I think is concise in the information, easy to read. I have three thoughts about your work. The first one is to have a better description of what has been found in relation to sleep habits in adults, since they only point out that there are few studies, but a better description would highlight their introduction. The second thing is abourt exercise habits. Same thing, I would like to know what tipe of exercise the older adults do. Is not explained in the data collection. It would be interesting. The last thing for discussion is to ask how to get more adults to exercise? It would be interesting know theirs opinions. |
||
|
Response 1: Thank you sincerely for your invaluable review and the meticulous suggestions you have provided. We deeply appreciate all of your recommendations. Below is our response: (1) We greatly appreciate your thorough review. In response, we have further summarized and addressed the deficiencies in the existing literature while identifying gaps in research concerning the relationship between consistent exercise routines and sleep quality among the elderly, as elaborated upon in lines 53 to 73. (2) We utilized the IPAQ questionnaire to categorize the exercise types of our interviewees. Regrettably, during the actual interview process, our interviewers discovered that certain activities such as the Eight Section Brocade, Five Animals of Qigong, and Tai Chi Chuan were not included in the questionnaire's data collection indicators. Our response to this issue is detailed in lines 313 to 320. (3) Within the discussion section, we have proposed the concept that the environment significantly influences the sleep quality of the elderly. Specifically, we suggest that environmental changes resulting from the COVID-19 pandemic, deemed a public health emergency, have led to acute sleep deprivation and alterations in the sleep quality of the elderly. These ideas are expounded upon in lines 303 to 306. |
||
|
|
||
|
|
||
|
|
||
|
|
||
|
|
||

Reviewer 4 Report
Thank you for the opportunity to review the manuscript titled: Unraveling the Nexus: Exploring the Relationship Between Exercise Habits and Sleep Quality in Older Adults
I could see that your study is interesting and you put a lot of effort into it.
But your manuscript needs some improvement.
1. The research design needs improvement.
2. The discussion lacks discussion of the main findings of this study. Please add further explanation to the discussion. For example, a physiological pathology may also explain the relationship between the two.
Author Response
Unraveling the Nexus: Exploring the Relationship Between Exercise Habits and Sleep Quality in Older Adults
|
Response to Reviewer 4 Comments
|
||
|
1. Summary |
|
|
|
Thank you very much for taking the time to review this manuscript. Please find the detailed responses below and the corresponding revisions in track changes in the re-submitted files. |
||
|
2. Questions for General Evaluation |
Reviewer’s Evaluation |
Response and Revisions |
|
Does the introduction provide sufficient background and include all relevant references? |
Can be improved |
We have added to the introduction. |
|
Are all the cited references relevant to the research? |
Can be improved |
We have broadened the existing literature by engaging in cross-regional discussions centered on our findings. |
|
Is the research design appropriate? |
Must be improved |
We supplemented the research design section. |
|
Are the methods adequately described? |
Can be improved |
We explain the shortcomings in the method description. |
|
Are the results clearly presented? |
Can be improved |
We expanded our findings. |
|
Are the conclusions supported by the results? |
Must be improved |
According to the results and discussion part, we supplement and modify the conclusion part to make it more consistent with the research results. |
|
3. Point-by-point response to Comments and Suggestions for Authors |
||
|
Comments 1: Thank you for the opportunity to review the manuscript titled: Unraveling the Nexus: Exploring the Relationship Between Exercise Habits and Sleep Quality in Older Adults. I could see that your study is interesting and you put a lot of effort into it. But your manuscript needs some improvement. 1. The research design needs improvement. 2. The discussion lacks discussion of the main findings of this study. Please add further explanation to the discussion. For example, a physiological pathology may also explain the relationship between the two. |
||
|
Response 1: Thank you sincerely for your invaluable review and the meticulous suggestions you have provided. We deeply appreciate all of your recommendations. Below is our response: (1) Based on the feedback from other reviewers and considerations of journal formatting, we have made revisions to our research design. For more comprehensive information, please refer to lines 87-149. (2) We express our gratitude for this valuable review, which has enabled us to enhance the elucidation of our main findings and provide additional insights in the discussion(lines 286-345). Moreover, we have incorporated insights from physiological pathology to attempt to elucidate the relationship between the two variables from the sight of environment.(lines 304-306). |
||

Reviewer 5 Report
Dear Authors,
Thank you for the opportunity to read your manuscript. This research provides an overview of the relationships between exercise habits and sleep quality in older adults. Overall, this article was well-organized and was interesting to read. I do have a couple of minor suggestions for your manuscript that I have included below. Wishing you all the best with your revisions.
Lines 43-45: This sentence sounds like something is missing. Maybe you mean “… to develop an implementation strategy for corresponding physical health intervention programs.”?
Line 54: Add a sentence to your justification clarifying what this study adds to the literature that maybe others have not. What is different between this study and the 3 previously published [references 13-15]?
Line 280: What were the limitations of your study?
Author Response
Unraveling the Nexus: Exploring the Relationship Between Exercise Habits and Sleep Quality in Older Adults
|
Response to Reviewer 5 Comments
|
||
|
1. Summary |
|
|
|
Thank you very much for taking the time to review this manuscript. Please find the detailed responses below and the corresponding revisions in track changes in the re-submitted files. |
||
|
2. Questions for General Evaluation |
Reviewer’s Evaluation |
Response and Revisions |
|
Does the introduction provide sufficient background and include all relevant references? |
Yes |
|
|
Are all the cited references relevant to the research? |
Yes |
|
|
Is the research design appropriate? |
Yes |
|
|
Are the methods adequately described? |
Yes |
|
|
Are the results clearly presented? |
Can be improved |
We expanded our findings. |
|
Are the conclusions supported by the results? |
Can be improved |
According to the results and discussion part, we supplement and modify the conclusion part to make it more consistent with the research results. |
|
3. Point-by-point response to Comments and Suggestions for Authors |
||
|
Comments 1: Thank you for the opportunity to read your manuscript. This research provides an overview of the relationships between exercise habits and sleep quality in older adults. Overall, this article was well-organized and was interesting to read. I do have a couple of minor suggestions for your manuscript that I have included below. Wishing you all the best with your revisions. (1) Lines 43-45: This sentence sounds like something is missing. Maybe you mean “… to develop an implementation strategy for corresponding physical health intervention programs.”? (2) Line 54: Add a sentence to your justification clarifying what this study adds to the literature that maybe others have not. What is different between this study and the 3 previously published [references 13-15]? (3) Line 280: What were the limitations of your study? |
||
|
Response 1: Thank you sincerely for your invaluable review and the meticulous suggestions you have provided. We deeply appreciate all of your recommendations. Below is our response: (1) Thank you for your attentive review. We have incorporated your comments into our revisions; please refer to lines 45-46 for specific details. (2) We greatly appreciate your thorough review. In response, we have further summarized and addressed the deficiencies in the existing literature while identifying gaps in research concerning the relationship between consistent exercise routines and sleep quality among the elderly, as elaborated upon in lines 53 to 73. (3) We have addressed research limitations within the discussion section. You can find a comprehensive discussion of the research limitations in this study from lines 286 to 345. |
||
|
|
||

Reviewer 6 Report
Dear authors,
The revision of the manuscript is in the attached document.
Kind regards.

Author Response
Unraveling the Nexus: Exploring the Relationship Between Exercise Habits and Sleep Quality in Older Adults
|
Response to Reviewer 6 Comments
|
||
|
1. Summary |
|
|
|
Thank you very much for taking the time to review this manuscript. Please find the detailed responses below and the corresponding revisions in track changes in the re-submitted files. |
||
|
2. Questions for General Evaluation |
Reviewer’s Evaluation |
Response and Revisions |
|
Does the introduction provide sufficient background and include all relevant references? |
Must be improved |
|
|
Are all the cited references relevant to the research? |
Must be improved |
We have broadened the existing literature by engaging in cross-regional discussions centered on our findings. |
|
Is the research design appropriate? |
Must be improved |
We supplemented the research design section. |
|
Are the methods adequately described? |
Must be improved |
We explain the shortcomings in the method description. |
|
Are the results clearly presented? |
Yes |
We expanded our findings. |
|
Are the conclusions supported by the results? |
Yes |
We have added to the introduction. |
|
3. Point-by-point response to Comments and Suggestions for Authors |
||
|
Comments 1: Dear authors, I have reviewed the manuscript entitled: "Unraveling the Nexus: Exploring the Relationship Between Exercise Habits and Sleep Quality in Older Adults". The aim of the study was to investigate the impact of exercise habits on improving the sleep quality of older adults while establishing a comprehensive exercise prescription program to optimize their sleep patterns. I would like to congratulate you on the research you have carried out, it is a very interesting study. After finalizing the revision of the manuscript, I would like to make a few comments: ABSTRACT 1) The length of the abstract is excessive based on the journal's instructions (it should be a maximum of 200 words). As an example, you could delete the sentences in lines 15-17 and 28-32.You can decide how to reduce the length. 2) The aim in lines 11-13 is not worded the same as in lines 57-60 and 255-256. The reader may be confused if it is not worded the same three times. Decide how to word the research aim and repeat it on all three occasions. KEYWORDS 3) The last word (investigation) should be deleted or changed, as it is very general and not related to the research topic. INTRODUCTION 4) I would like to congratulate you on the quality of this section. Despite the congratulations, the length of the section is too small. This section should be significantly expanded based on the scientific literature and the research topic. 5) Line 6 mentions "our country". This text should be changed to "China". METHODS 6) It would be appropriate to modify the structure of this section. There is information that does not seem to be well placed in a certain section. I propose the following structure: 2.1Design, 2.2 Subjects, 2.3 Instruments and variables, 2.4 Procedure, 2.5 Data analysis. 7) The type of study design is still to be identified. For example, cross-sectional studio. 8) The information on lines 69-70 ("Therefore, this study will draw on past research results")is somewhat confusing. The reader may misunderstand that it is a systematic review when it is not. These lines can be deleted to avoid confusion. 9) The information provided on the place of residence of the participants may lead to confusion. It should be reworded to make it easier to read. The relationship between districts H and Y should be explained concretely, as for example in the objective (line 15) District H is mentioned but in the results District Y is mentioned (line 252). 10) The population size should be stated. This would help to know how many people in the district are over 60 (only 18.46% are said to be over 60). 11) Other details of the sample calculation would need to be detailed to ensure that it has been done correctly, e.g. the assumed sampling error. 12) Line 96 mentions that 7 points have been used as the cut-off point. It would be useful to explain the reason and/or include other studies that have used the same cut-off point. 13) In other sections of the manuscript it is mentioned that participants take part in a healthy exercise promotion program. Details of this program should be given in the Methods section. RESULTS 14) I would like to congratulate you on the section you have produced. DISCUSSION 15) The first paragraph of the section should contain the research aim as mentioned in other sections. 16) In lines 257-258 it is mentioned that the key conclusions will be presented. Change the wording of these lines because this is the discussion section, not the conclusions section. 17) The discussion section is too short. The length and quality of the section should be increased. To this end, the information provided should be expanded on the basis of the scientific literature, inferences should be made as to why it is believed that these results may have been obtained, the limitations of the study should be mentioned and future lines of research should be described. CONCLUSIONS 18) I would like to congratulate the authors for the way they have prepared this section. REFERENCES 19) The way in which references are cited is not the way commonly used in Healthcare journal publications. It should be adapted. n addition, the number of references should be increased, as it is very low (only 35). I would like to end by again congratulating the authors of the manuscript on the work they have done. I hope you will take all my comments into account |
||
|
Response 1: We sincerely appreciate your invaluable comments and thorough suggestions. Your input has been of great significance to us. Following a thorough discussion, we wholeheartedly concur with all of your suggestions and have implemented the necessary corrections accordingly. Please find our response below: (1) Thank you for your reminder, we have condensed the summary after integrating your comments. (2) We have unified the purpose of the three scenarios and examined the relevant statements in the conclusion. (3) Thanks for your detailed suggestion, we have deleted it. (4) Incorporating your feedback and drawing upon relevant publications within this journal, we have meticulously reviewed the existing literature and identified its limitations in the introduction section. Subsequently, we have undertaken substantial expansion and enhancement of this section in alignment with the research focus.(lines 44 to 64) (5) We apologize for making such an obvious mistake and we have corrected it. (6) We have revised the METHODS part based on your comments and with reference to the format of other articles published in this journal. (7) We have clearly defined it in line 79. (8) We have deleted these lines. (9) We have corrected this problem and changed the Y district into Y Street in line 90. (10) We have added this information in lines 82 to 84. (11) We update this information on lines 89-90. (12) Following the introduction of the PSQI questionnaire in China, several scholars conducted reliability and validity tests to assess its applicability. Subsequently, in Chinese research, a score of 7 has been employed as a consistent threshold to categorize the sleep status of Chinese respondents. This standard continues to be utilized in recent Chinese studies, as elaborated in lines 123 to 125. (13) Regrettably, the health exercise promotion program in which the project team participants were involved was not organized in a strictly categorized manner. Furthermore, the subjects under investigation in this study did not align with those participating in the health exercise promotion program, as only the Tai Chi project was implemented. The selection of elderly participants for this study was carried out through stratified random sampling. Therefore, we have chosen not to include a detailed description of this health exercise promotion program within the manuscript. (14) We appreciate your comments. (15) We also restated the purpose of the study at the beginning of the discussion, as detailed in lines 277-280. (16) We have revised the statement about "conclusions", which can be seen in lines 277-282. (17) We express our gratitude for this valuable review, which has enabled us to enhance the elucidation of our main findings and provide additional insights in the discussion(lines 286-345). (18) We appreciate your comments. (19) We have revised the reference formatting in accordance with the guidelines established by the Healthcare journal. Additionally, we have expanded the references based on the feedback received from all reviewers. |
||
|
|
||
|
|
||

Round 2
Reviewer 2 Report
I am pleased with the changes. I would like to thank the authors.
Author Response
Thanks
Reviewer 6 Report
Dear Authors,
I have reviewed again the manuscript entitled: "Unraveling the Nexus: Exploring the Relationship between Exercise Habits and Sleep Quality in Older Adults".
I want to thank you for taking most of my comments into consideration. In my view, the quality of the manuscript has improved substantially.
One of the comments I made in the previous review (issue 19) was that you had to adapt the citation of references to the requirements of the Healthcare journal. This comment has not been taken into consideration.
Best regards.
Author Response
Thank you, we have already modified the format of the references.